# Circulating miR-200 Family and CTCs in Metastatic Breast Cancer before, during, and after a New Line of Systemic Treatment

**DOI:** 10.3390/ijms23179535

**Published:** 2022-08-23

**Authors:** Chiara Fischer, Andrey Turchinovich, Manuel Feisst, Fabian Riedel, Kathrin Haßdenteufel, Philipp Scharli, Andreas D. Hartkopf, Sara Y. Brucker, Laura Michel, Barbara Burwinkel, Andreas Schneeweiss, Markus Wallwiener, Thomas M. Deutsch

**Affiliations:** 1Department of Obstetrics and Gynecology, University of Heidelberg, Im Neuenheimer Feld 440, 69120 Heidelberg, Germany; 2German Cancer Research Center (DKFZ), Im Neuenheimer Feld 280, 69120 Heidelberg, Germany; 3Institute of Medical Biometry, University of Heidelberg, Im Neuenheimer Feld 130.3, 69120 Heidelberg, Germany; 4Department of Obstetrics and Gynecology, University of Tübingen, Calwerstraße 7, 72076 Tuebingen, Germany; 5National Center for Tumor Diseases, Im Neuenheimer Feld 460, 69120 Heidelberg, Germany

**Keywords:** circulating microRNA, microRNA-200 family, miR-200 family, miR-200s, circulating tumor cells, CTC, metastatic breast cancer, MBC, epithelial-mesenchymal transition, EMT

## Abstract

The extracellular circulating microRNA (miR)-200 regulates epithelial-mesenchymal transition and, thus, plays an essential role in the metastatic cascade and has shown itself to be a promising prognostic and predictive biomarker in metastatic breast cancer (MBC). Expression levels of the plasma miR-200 family were analyzed in relationship to systemic treatment, circulating tumor cells (CTC) count, progression-free survival (PFS), and overall survival (OS). Expression of miR-200a, miR-200b, miR-200c, miR-141, and miR-429, and CTC status (CTC-positive ≥ 5 CTC/7.5 mL) was assessed in 47 patients at baseline (BL), after the first completed cycle of a new line of systemic therapy (1C), and upon the progression of disease (PD). MiR-200a, miR-200b, and miR-141 expression was reduced at 1C compared to BL. Upon PD, all miR-200s were upregulated compared to 1C. At all timepoints, the levels of miR-200s were elevated in CTC-positive versus CTC-negative patients. Further, heightened miR-200s expression and positive CTC status were associated with poorer OS at BL and 1C. In MBC patients, circulating miR-200 family members decreased after one cycle of a new line of systemic therapy, were elevated during PD, and were indicative of CTC status. Notably, increased levels of miR-200s and elevated CTC count correlated with poorer OS and PFS. As such, both are promising biomarkers for optimizing the clinical management of MBC.

## 1. Introduction

Despite tremendous diagnostic and therapeutic improvement, female breast cancer continues to prevail as the fifth leading cause of cancer mortality, with 2.3 million new cases and 68,500 deaths worldwide [1]. In the era of precision oncology and molecular profiling of individual patients’ tumors, factors that drive tumor metastasis and biomarkers of diagnostic and prognostic value have been the subject of intense scrutiny and research [2]. Readily available and minimally invasive, tumor-derived blood components, such as circulating miRNA, circulating tumor cells (CTCs), cell-free DNA, circulating tumor DNA, circulating tumor RNA, exosome vesicles, and tumor-educated platelets, have attracted an extraordinary degree of scientific attention [3,4].

CTC enumeration, especially over time, is a known independent prognostic marker for unfavorable overall survival (OS) and progression-free survival (PFS) in metastatic breast cancer (MBC). Furthermore, CTCs could offer enticing and non-invasive tools to evaluate real-time molecular genomic characteristics, tumor heterogeneity and tumor alterations during treatment. Congruently, an increasing body of research currently investigates treatment choice and clinical monitoring based on dynamic CTC evaluation, including early response to systemic therapy [5,6,7,8,9,10]. In this regard, Bidard et al. recently proposed CTC count as a decisive biomarker for selecting first-line treatment in hormone receptor-positive HER2-negative MBC [11]. However, a routine clinical use case regarding diagnostic and therapeutic strategies remains to be defined [12,13,14]. Apart from their clinical application, CTCs have also been shown to play an essential supporting role in the metastatic cascade at the pathophysiological level [15,16,17].

MicroRNAs (miRNAs) are short, non-coding RNA sequences that regulate the expression of multiple genes at post-transcription level. In addition, remarkably stable cell-free (or extracellular) miRNAs have been found in biological fluids, being either associated with proteins of the Argonaute family or within certain extracellular vesicles [18]. The well-studied miR-200 family (miR-200a, miR-200b, miR-200c, miR-141, and miR-429) regulates an essential process in the metastatic cascade called epithelial-mesenchymal transition (EMT) [19]. In the early phases of the metastatic process, low levels of miR-200s enhance EMT and enable tumor cells to acquire motile mesenchymal cell phenotypes via induction of vimentin expression, allowing them to migrate and invade surrounding tissues. However, in later stages of metastatic development, high miR-200 levels facilitate mesenchymal-epithelial transition (MET) by enabling cells to acquire an epithelial cell phenotype via induction of E-cadherin expression, allowing them to successfully establish in infiltrated organs [20,21,22]. Furthermore, all members of the miR-200 family were of significant prognostic value regarding PFS and OS [23,24,25]. Clinically intriguing, Shao et al. demonstrated elevated expression of miR-200a in chemotherapy-resistant patients compared to chemo-sensitive patients [26]. As miRNAs influence entire biological pathways, rather than single genes, miRNA mimics and miRNA inhibitors, respectively, are being developed for therapeutic purposes, especially regarding immune checkpoint therapy [19,27,28,29]. Moreover, miRNA and CTC status correlated positively with serum levels of miR-200a, miR-200b, miR-200c, and miR-141 and number of CTCs (≥5 CTC/7.5 mL blood) in MBC. Additionally, Madhavan et al. showed that the prognostic value of miR-200b was equivalent to, or even better than, CTC status. Hence, paired analysis of miR-200b and CTC status significantly outperforms individual assessment [23].

In this study we evaluated levels of circulating EMT-related miR-200 members in blood plasma over the course of MBC and assessed the capacity of these miRNAs to serve as clinical biomarkers for MBC, along with the corresponding CTC status.

## 2. Results

### 2.1. Patient Characteristics

The present study enrolled 47 patients with MBC. Patient and tumor characteristics have been reported in a previous article and can be found in the Appendix A) [30]. At study entry, on average, participants were 57 years old (Range: 35–89). Patients’ receptor status of estrogen was positive in 38 patients (80.8%), receptor status of progesterone was positive in 32 patients (68.1%), and of human epidermal growth factor receptor 2 (HER2) was positive in 5 patients (10.6%), respectively. Further, 12 patients (25.5%) presented with a single distant site of metastasis, 15 (31.9%) with two, and 20 (42.6%) with more than 3, respectively (Appendix A).

### 2.2. MiRNA Expression across Time

Figure 1 illustrates expression levels of circulating miR-200a, miR-200b, miR-200c, miR-141, and miR-429 at BL, 1C and PD. Cp-values and miRNA expression are to be read inversely, meaning a higher Cp value corresponds to lower miRNA expression, and conversely. Mean detection rates of the measured miRNA panel are listed in the Appendix A).

Linear regression analysis supported a significant association of all miR-200s and CTC status. A significant relation was also found between miR-200a and miR-200b expression and local metastasis and miR-200s and bone metastasis, respectively (Appendix A).

Expression levels of miR-200a, miR-200b, and miR-141 at BL were significantly higher compared to respective levels at 1C (*p* ≤ 0.03, Table 1). Furthermore, expression levels of all miR-200_PD_ were higher than miR-200_1C_ (*p* ≤ 0.004, Table 1). There were no significant differences in any plasma miRNA_BL_ expression patterns compared to miRNA_PD_ (Figure 1, Table 1).

### 2.3. CTC Status

CTC status evaluation showed 26 (55.3%) CTC_BL_-positive vs. 21 (44.7%) CTC_BL_-negative patients, 19 (40.4%) CTC_1C_-positive vs. 28 (59.6%) CTC_1C_-negative patients, and 24 (51.1%) CTC_PD_-positive vs. 23 (48.9%) CTC_PD_-negative patients. 

CTC status switched from CTC_BL_-negative to CTC_1C_-positive in 4 (8.4%) patients, from CTC_BL_-positive to CTC_1C_-negative in 11 (23.4%) patients and remained the same in 32 (68.1%) patients. At PD, CTC status changed from CTC_1C_-negative to CTC_PD_-positive in 8 (17.0%) patients, from CTC_1C_-positive to CTC_PD_-negative in 3 (6.4%) patients and remained the same in 36 (76.6%) patients (Table 2).

The expression levels of all miR-200s investigated were predictors of CTC status and significantly higher among the CTC-positive than the CTC-negative patient samples at all observed time points (Table 3 and Appendix A).

### 2.4. MiRNA, CTC and Survival

Median OS was 21 months (Range: 3–84). At BL, log-rank analysis showed a significant reduction in OS for patients presenting increased expression of miR-200a, miR-200b, miR-200c, miR-141, and miR-429, respectively (*p* ≤ 0.009, Table 4). Further, a positive CTC_BL_ status correlated with poorer OS compared to patients with negative CTC_BL_ status (*p*_CTC_ < 0.001). These findings were identical at 1C, namely, heightened miRNA_1C_ expression levels and elevated CTC_1C_ count correlated with unfavorable OS (*p* ≤ 0.002). At PD, however, only elevated miR-429_PD_ levels and positive CTC_PD_-status correlated with reduced OS (*p*_miR-429_ = 0.032, *p*_CTC_ < 0.001, Table 4).

Regarding PFS, no significant association between any miRNA_BL_ or CTC_BL_ status was observed at baseline (Table 5). However, all members of the miR-200_1C_ family and CTC_1C_ status significantly correlated with unfavorable PFS (*p*_miR-200a_ = 0.028, *p*_miR-200b_ = 0.037, *p*_miR-200c_ = 0.005, *p*_miR-141_ = 0.001, *p*_miR-429_ = 0.021, *p*_CTC_ = 0.011). Accordingly, cox regression analysis confirmed the prognostic association of miR-200s and CTC regarding OS and PFS (Appendix A).

## 3. Discussion

This study found that serum levels of circulating EMT-related miR-200s were increased before therapy (miR-200a, miR-200b, miR-141) and at the progression of disease (miR-200a, miR-200b, miR-200c, miR-141, and miR-429) when compared to levels after 1 cycle of systemic therapy. These findings align with prior reports showing elevated expression levels of miR-200c and miR-141 during tumor progression and metastasis [19,31]. Additionally, elevated miR-200b and miR-200c serum levels accurately discriminated between patients suffering from early-stage breast cancer and MBC. Interestingly, the results showed increased miR-200b serum levels in premenopausal compared to postmenopausal MBC patients [25]. The present results align with current literature indicating that all circulating miR-200s are upregulated during metastatic disease.

Moreover, studies suggested an active regulatory role of circulating miRNAs regarding metastatic processes in vivo. This mode of action may include transferring metastatic capabilities to non-metastatic cells via extracellular vesicles containing miR-200 [32]. Dykxhoorn and colleagues found that injection of tumor cells overexpressing miR-200s resulted in formation of macroscopic lung and liver metastasis in mice [20]. In another mouse xenograft model, overexpression of miR-200c and miR-141 mediated by SerpinB2 led to metastasis of CTCs in lung and lymph nodes [33]. Furthermore, higher blood levels of miR-200b were found in lymph node-positive compared to lymph node-negative breast cancer patients, and in patients with distant metastasis compared to patients without metastases [34]. Additionally, miR-200a was found to protect cancer cells from anoikis and was associated with lymph node metastasis [35]. Since levels of all miR-200 family members were significantly increased during tumor progression compared to 1C, said miRNAs may actively promote cancer progression.

To date, the true origin of circulating miRNA is not fully understood. Several hypotheses suggest passive release by intact, apoptotic or necrotic circulating cells of the primary or metastatic tumor. In contrast, others propose active secretion in extracellular vesicles by blood cells or CTCs [36,37,38]. By showing increased miRNA levels during tumor progression and a significant correlation with the number of intact CTCs, the results of this study lead to a proposition wherein CTCs that have been shed from a metastatic site are postulated as primary origin of the circulating miR-200 family. Further research is required in order to advance the aforementioned theories, as neither have led to a paradigm shift.

Madhavan et al. documented that CTC-positive and CTC-negative patients differed significantly regarding the above-mentioned miR-200s [23]. Accordingly, Markou et al. reported increased circulating miR-200c expression in CTC-positive MBC patient samples [36]. The present findings replicated said reports by showing a strong correlation and significant predictive value concerning elevated miR-200s expression and increased CTC count. Notably, said correlation persisted despite systemic therapeutic intervention, and, in addition to the panel investigated by Madhaven et al., applied to miR-429 as well [23].

Regarding prognostic capacities, at BL, increased miR-200a, miR-200b, miR-200c, and miR-141 expression was associated with poorer OS, supporting existing findings [22,23,39]. At 1C, all miR-200s were significantly associated with reduced OS. Interestingly, elevated miR-429 levels persisted in indicating an unfavorable outcome at PD.

Further, increased miR-200s were associated with shortened PFS at 1C. However, and in contrast to results by Madhavan et al., no such relation was observed at BL [23]. Circumstances leading to inconsistent results could include small cohort size, differences in disease severity among patients recruited in comparable studies, differences in inclusion and exclusion criteria regarding patients’ biological characteristics, and differences in sampling techniques and laboratory protocols.

In line with existing knowledge, increased CTC enumeration (≥5 CTC/7.5 mL) was associated with poorer OS at BL, 1C, and PD, thereby supporting their robust prognostic capacities [12,40,41,42]. However, contradicting prior studies, there was no significant association of CTCs and reduced PFS at BL [6]. Factors possibly influencing such a lack of reproducibility are listed above.

In conclusion, miR-200s, as well as CTC count, are valuable prognostic biomarkers for clinical management in MBC.

The overall clinical goal is to effectively monitor tumor burden and therapeutic efficacy. In said pursuit, Shao et al. reported elevated serum levels of miR-200a as predictive of chemotherapy resistance in MBC, irrespective of treatment regimens [26]. Furthermore, miR-200b was associated with response to docetaxel in prostate and radiotherapy in non-small-cell lung cancer patients [43,44]. This study showed significantly lower expression of miR-200a, miR-200b, and miR-141 at 1C compared to BL, possibly indicating that systemic therapy contained disease-related miRNA expression patterns. Interestingly, the number of intact and apoptotic CTCs at BL and 1C were also reliable indicators of response to therapy [5,6,9,41]. Accordingly, yet controversially, the AGO guideline recommended counting CTCs during the first three weeks of systemic therapy could be used to assess early response to treatment in breast cancer patients [45,46,47].

Additionally, miRNAs are also considered potential therapeutic targets [48]. For example, by modulating and regulating various signaling pathways within the tumor microenvironment, such as tumor cell PD-L1 checkpoint expression by the miR-200/ZEB1 axis, miRNAs support or inhibit specific mechanisms from negatively affecting tumorigenesis. Therefore, miRNAs could be of interest for future immunotherapeutic strategies, including immune checkpoint blockade, adoptive cell therapy, cancer vaccines, and cytokine therapy [19,29,49,50,51].

A major limitation of this study is the small sample size, which must be considered when interpreting the results. Further, the FDA-approved CTC CellSearch^®^ system, while commonly used, is not very sensitive in measuring CTCs that have undergone EMT due to its EpCAM-based detection mechanism. Newer approaches include fluorescent flow cytometry and in vivo optical and photoacoustic imaging technology. In general, studies of circulating miRNAs are challenged by the lack of a standardized normalization method for Cp values of miRNAs in serum. The respective quality limits have to be considered [52]. Finally, Cp values above 35 must be interpreted with caution as they may indicate inefficient RNA extraction. Although miRNA detection by RT-qPCR is highly sensitive, its multiplex capacity is limited, due to the lack of spectrally resolved fluorescent probes. Among the new innovative analytical methods, suspension arrays are promising high-throughput and multiplexable analytical techniques with high sensitivity and compatibility with multiple readout options. Sensitive readout approaches include SERS (surface-enhanced Raman scattering), SPR (surface plasmon resonance), and electrochemical labeling (e.g., CRISPR-based biosensing platforms) [53]. Excitingly, miRNAs could recently be measured directly in CTCs in vivo, promising great tumor specific diagnostic, prognostic, and therapeutic potential [54,55]. However, RT-qPCR is undisputedly considered the gold standard for miRNA detection, as it offers attomolar sensitivity and specificity for single cells and is also cost-effective [53].

Thus, future research is encouraged to further explore pathophysiological relationships, prognostic and predictive values and further therapeutic use of circulating miRNA biomarkers.

## 4. Materials and Methods

### 4.1. Study Design and Samples

The present analysis was a retrospective, single-center, cohort study conducted at the National Center for Tumor Diseases, Heidelberg, Germany, in conjunction with the German Cancer Research Center (DKFZ), Heidelberg, Germany, and the Department of Obstetrics and Gynecology, University of Heidelberg, Heidelberg, Germany. Ethical approval was obtained by the ethics committee of the Medical Faculty Heidelberg of the University of Heidelberg, approval No. S-295/2009. All patients provided written informed consent. Enrolled were female MBC patients ≥ 18 years starting a new line of systemic therapy between April 2010 and September 2011. Previous treatment and time of initial diagnosis were disregarded. Exclusion criteria were repeated inclusion, retracted consent, missing follow-up, male sex, and missing clinical data or unavailable blood samples. Tumor burden was assessed at study entry and three-month intervals according to the RECIST criteria (Response Evaluation Criteria in Solid Tumors) using radiological imaging [56]. Survival (OS, PFS) was calculated as time, in months, from first blood draw at enrolment until disease progression, death, or loss to follow-up. Median follow-up was 21.5 months (Interquartile Range 23.3 months). Clinical and histopathological characteristics were collected retrospectively from the medical records provided by University Hospital Heidelberg and, in some instances, by peripheral hospitals without re-evaluation.

### 4.2. MiRNA Assessment

Blood plasma sampling, cell-free RNA isolation as well as cell-free miRNA detection were performed as described previously [30]. The expression levels of miR-200a, miR-200b, miR-200c, miR-141, and miR-429 were measured, respectively, at the three following consecutive time points in the plasma of 47 patients with MBC: (1) at baseline before the start of a new line of systemic therapy (BL, miRNA_BL_), (2) after the first completed cycle of a new line of systemic therapy (1C, miRNA_1C_), and (3) at the radiologically confirmed progression of the disease (PD, miRNA_PD_).

#### 4.2.1. Isolation of Circulating miRNAs

For sampling, 7.5 mL of peripheral blood were collected in 9 mL EDTA tubes (Sarstedt S-Monovette^®^, Nuernbrecht, Germany) and processed within 2 h, following a two-step centrifugation protocol: 1300× *g* for 20 min at 10 °C, followed by 15,500× *g* for 10 min at 10 °C. Then, samples were snap-frozen and stored at −80 °C. TRI-Reagent LS^®^ (Sigma-Aldrich, St. Louis, MO, USA) and Qiagen miRNeasy^®^ mini kit (Qiagen, Hilden, Germany) were used to extract miRNA from plasma, as described in Turchinovich et al. [57].

All laboratory processes were performed at room temperature. Firstly, 400 µL blood plasma was denatured in 2 mL Eppendorf tubes by adding 1200 µL TRI-Reagent LS^®^. Subsequently, 1 pg synthetic cel-miR-39 (spiked-in normalization reference) and 1 µL glycogen (10 mg/mL) (Thermo Scientific, Life Technologies, Carlsbad, CA, USA) were added. Then, samples were vortexed for 15 s and incubated for 20 min. After adding 220 µL chloroform, samples were vortexed for 15 s, incubated for 5 min, and finally centrifuged at 16,000× *g* for 20 min. The 600 µL of aqueous supernatant containing dissolved total RNA was collected, mixed with 900 µL RNase-free 100% ethanol (Roth Chemicals, Karlsruhe, Germany), and incubated for 5 min. According to the manufacturer’s instructions, the total RNA was then purified by miRNeasy^®^ kit (Qiagen, Hilden, Germany). Finally, extracted RNA was eluted in 60 µL RNase-free water and stored at −80 °C until further processing.

#### 4.2.2. Quantitative Real-Time PCR

The current analysis is based on a panel of 13 miRNAs developed and validated within a previously published study by Madhavan et al. [23]. The following miRNAs were analyzed: miR-16, miR-24, miR-29a, miR-138-5p, miR-141, miR-200a, miR-200b, miR-200c, miR-210, miR-375, miR-365, miR-429, and miR-1260. Of this panel, the miR-200 family (miR-200a, miR-200b, miR-200c, miR-141, and miR-429) was of particular interest for this study due to its role in the EMT process.

MiRNA expression levels and the exogenous spike-in normalizer cel-miR-39 were analyzed using the TaqMan^®^ miRNA Assays (Applied Biosystems, Life Technologies, Carlsbad, CA, USA). RT reactions comprised 5 µL of previously purified miRNA sample, 1 µL each of stem-loop RT primers, 1.5 µL of RT buffer 10×, 0.15 µL of dNTPs 100 mM, 0.19 µL of RT inhibitor, 1 µL of Multiscribe RTase 50 U/µL, and RNase-free water.

RT reactions had a final volume of 15 µL and were subsequently incubated on 96-well plates for 30 min at 16 °C, followed by 30 min at 42 °C, 5 min at 85 °C, and held at 4 °C. Samples were diluted in RNase-free water and stored at −20 °C.

Real-time qPCR was performed using TaqMan^®^ Universal PCR Master Mix (Applied Biosystems, Foster City, CA, USA). A 10µL PCR reaction included 2 µL of diluted RT product and 1× TaqMan^®^ Universal PCR Master Mix and 1× of each of the miRNA assay primers. Reactions were incubated in 384-well plates at 95 °C for 10 min, followed by 50 cycles of 95 °C for 50 s each, and 60 °C for 1 min. Samples were analyzed in duplicate. Real-time PCR was performed using LightCycler^®^ 480 Real-Time PCR System, and data were analyzed with the LightCycler^®^ 480 software (Roche Diagnostics, Mannheim, Germany). Crossing point (Cp), calculated by determining the second derivative maxima of the amplification method, was given as output. MiRNAs not detected were filtered out.

Regarding miRNAs measured with TaqMan^®^ Assay, the exogenously added cel-miR-39 was measured simultaneously with other miRNAs and used as a normalizer for RT-qPCR data. The normalization factor was calculated using the difference between the median cel-miR-39 Cp values of all samples and the mean cel-miR-39 Cp values of a respective patient sample. Cp values are inversely related to one another, in that a higher Cp value indicates lower miRNA expression, and vice versa.

### 4.3. CTC Assessment

CTC count was evaluated in 7.5 mL peripheral blood using the Cell-Search^®^ Assay (CellSearch^®^ and Epithelial CellKit/CellSpotter^®^, Janssen Diagnostics, LLC, Raritan, NJ, USA). After collecting the blood in CellSave^®^ tubes, it was stored at room temperature and processed within 96 h. Samples were prepared and processed according to the manufacturer’s instructions. Trained personnel conducted the CTC analysis, and independent reviewers confirmed the results. Blood samples with ≥5 CTC/7.5 mL were considered CTC-positive and samples with less than 5 CTC/7.5 mL blood as CTC-negative, respectively [58]. Staff involved in CTC analysis and/or study management were blinded to patients’ clinical data.

### 4.4. Statistical Analysis

Clinical characteristics and CTC status are described as absolute and relative frequencies for binary and ordinal variables and as median, mean, and range for continuous variables. A linear regression model was calculated to assess CTC status as well as miRNA expression levels in groups with each of the following characteristics: patients’ age at initial diagnosis and baseline, tumor biology, hormone receptor status, site of metastasis, therapy, and CTC status. MiRNA detection rates were presented as mean Cp value and standard deviation. Boxplots and Pairwise Student’s *t*-tests were conducted to display expression levels and test for overall expression differences between chosen time points, respectively. The association and predictive value of miRNA expression in relation to CTC-status was analyzed using Student’s t-test and logistic regression analysis, respectively. Log-rank tests were calculated for survival analysis (OS, PFS). The Cox proportional hazard regression model was used to analyze potential bias factors, including the following: age at initial diagnosis, distant metastasis at initial diagnosis, local, visceral, and bone metastasis.

Statistical analysis was performed using R version 3.5.1 (2 July 2018) [59]. The significance level was set at alpha 5% and interpreted descriptively (exploratory study).

## 5. Conclusions

In patients with MBC, expression levels of circulating miR-200a, miR-200b, miR-200c, miR-429, and miR-141 were significantly elevated during disease progression and predictive of CTC status. Both elevated CTCs and increased circulating miR-200s content in blood plasma were associated with reduced OS and PFS.

As an adjunct component of a reliable, valid, and minimally invasive liquid biopsy, miRNAs and CTCs may valuably contribute to individualized MBC management.

## Figures and Tables

**Figure 1 ijms-23-09535-f001:**
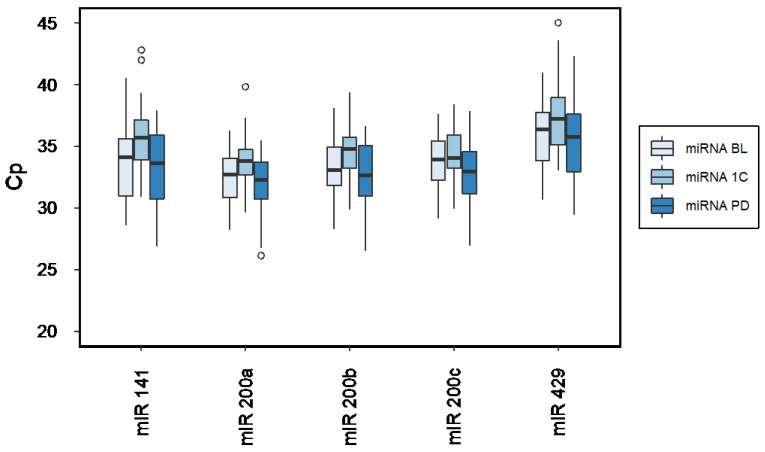
Comparison of miR-200s expression levels at baseline (miRNA_BL_), after 1 completed cycle of a new line of systemic therapy (miRNA_1C_), and at the progression of disease (miRNA_PD_) (outliers are shown as circles).

**Table 1 ijms-23-09535-t001:** Results of a pairwise t-test comparing the miRNA expression levels across measurements in time.

	BL vs. 1C	1C vs. PD	BL vs. PD
miRNA	*p*	*p*	*p*
miR-200a	0.032	0.002	0.964
miR-200b	0.028	0.001	0.689
miR-200c	0.459	0.003	0.167
miR-141	0.003	0.001	1.00
miR-429	0.076	0.004	0.827

BL—baseline, 1C—after 1 completed cycle of a new line of systemic therapy, PD—progression of disease.

**Table 2 ijms-23-09535-t002:** CTC status and CTC status changes of patients across measurements in time.

	Positive		Negative		CTC Pos to CTC Neg		CTC Neg to CTC Pos		Stable	
	N	%	N	%	N	%	N	%	N	%
CTC_BL_	26	(55.3)	21	(44.7)	-	-	-	-	-	-
CTC_1C_	19	(40.4)	28	(59.6)	11	(23.4)	4	(8.4)	32	(68.1)
CTC_PD_	24	(51.1)	23	(48.9)	3	(6.4)	8	(17.0)	36	(76.6)

CTC—circulating tumor cells, Positive—≥5 CTC/7.5 mL blood, Negative—<5 CTC/7.5 mL blood, N—sample size, CTC_BL_—CTC status at baseline, CTC_1C_—CTC status after 1 completed cycle of a new line of systemic therapy, CTC_PD_—CTC status at progression of disease.

**Table 3 ijms-23-09535-t003:** Results of Student’s *t*-test comparing miRNA expression levels in patients with positive and negative CTC status.

	CTC_BL_	CTC_1C_	CTC_PD_
miRNA	*p*	*p*	*p*
miR-200a	<0.001	<0.001	0.003
miR-200b	<0.001	0.001	0.009
miR-200c	0.001	0.001	0.014
miR-141	<0.001	0.001	0.010
miR-429	<0.001	0.001	0.007

CTC—circulating tumor cells, CTC_BL_—CTC status at baseline, CTC_1C_—CTC status after 1 completed cycle of a new line of systemic therapy, CTC_PD_—CTC status at progression of disease, *p*—statistical *p*-value.

**Table 4 ijms-23-09535-t004:** Log-rank test comparing OS distribution among levels of miRNA expression and CTC status.

	BL			1C			PD		
		Median OS (Months)			Median OS (Months)			Median OS (Months)	
miRNA	*p*	miRNA High	miRNA Low	*p*	miRNA High	miRNA Low	*p*	miRNA High	miRNA Low
miR-200a	0.006	14	26	0.001	11	24	0.067	9	14.5
miR-200b	0.006	14	26	0.001	11	24	0.167	9	14
miR-200c	0.007	15	26	<0.001	12	26	0.102	9	14
miR-141	0.009	14	26	<0.001	11	26	0.130	10	14.5
miR-429	0.003	12	26	0.002	11	23	0.032	7	14.5
		CTC-pos	CTC-neg		CTC-pos	CTC-neg		CTC-pos	CTC-neg
CTC	<0.001	16	31	<0.001	14	25	<0.001	8	19

Samples dichotomized as lower quartile (miRNA high) and upper rest (miRNA low) based on their Cp values, or based on their CTC status. CTC—circulating tumor cells, OS—overall survival, CTC-pos—≥5 CTC/7.5 mL, CTC-neg—<5 CTC/7.5 mL, BL—baseline, 1C—after 1 completed cycle of a new line of systemic therapy, PD—progression of disease, *p*—statistical *p*-value.

**Table 5 ijms-23-09535-t005:** Log-rank test comparing PFS distribution among miRNA expression and CTC status.

	BL			1C		
	*p*	Median PFS (Months)		*p*	Median PFS (Months)	
miRNA		miRNA High	miRNA Low		miRNA High	miRNA Low
miR-200a	0.302	3.5	6	0.028	1.5	4
miR-200b	0.302	3.5	6	0.037	1.5	4
miR-200c	0.118	3	6	0.005	1.5	4
miR-141	0.202	4	5	0.001	1.5	4
miR-429	0.346	3.5	7	0.021	1	4
		CTC pos	CTC neg		CTC pos	CTC neg
CTC status	0.238	3.5	9	0.011	2	4

Samples dichotomized as lower quartile (miRNA high) and upper rest (miRNA low) based on their Cp values or based on their CTC status. CTC—circulating tumor cells, PFS—progression-free survival, CTC-pos—≥5 CTC/7.5 mL, CTC-neg—<5 CTC/7.5 mL, BL—baseline, 1C—after 1 completed cycle of a new line of systemic therapy, *p*—statistical *p*-value.

## Data Availability

The data sets generated and/or analyzed in the current study are available from the corresponding author upon reasonable request.

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
