# Peer review of "Circulating miR-200 Family and CTCs in Metastatic Breast Cancer before, during, and after a New Line of Systemic Treatment"

_ijms, 2022, doi:10.3390/ijms23179535_

Round 1
Reviewer 1 Report
Dear authors, I have reviewed your excellent manuscript entitled: ‘Circulating miR-200 Family and CTCs in Metastatic Breast Cancer before, during, and after a New Line of Systemic Treatment’ and I have the following comments:
Minor points:
1. In abstract all abbreviations should be explained.
2. Please specify (paragraph 4.1.) the median and interquartile range of follow-up.
3. Please provide inclusion and exclusion criteria.
4. The study population is quite small thus I suggest to modify conclusions into more speculative ones.
5. The authors should provide limitations of the study.
Author Response
Thank you for the constructive criticism. We have tried to implement them as best as possible in our manuscript.

Reviewer 2 Report
This review is on “Circulating miR-200 Family and CTCs in Metastatic Breast Cancer before, during, and after a New Line of Systemic Treatment” by Fischer and colleagues. In a retrospective analysis the authors investigated a rather small cohort of 47 metastatic breast cancer patient from a single-center, starting a new line of systemic therapy in 2010/2011, i.e. over a decade ago.
The authors compare circulating levels of miR-200 family members (miR-200a, miR-200b, miR-200c, miR-141, and miR-429) as well as CTCs with tumor progression, and how this could be used as a valuable and independent prognostic marker for PFS and OS.
Optically, although zoomed in and with the x-axis starting at 20 (not at 0, but that is OK), Figure 1 shows only a rather small effect, but it is consistent with all five miRs. Statistically the authors confirm the effects claimed. The miR-200s also predicted the CTC status.
Both, high miR-200s and CTCs correlate with reduced PFS and OS. The study is consistent with similar findings, but in principle the reviewer likes the investigation of CTCs in parallel to the miRs.
Although the reviewer is not fully convinced of all the aspects and views of the study, the reviewer cannot detect any major mistakes. The same cohort of 47patients was also published recently by almost the same group of authors in different order, but with a different focus and not highly overlapping results, so this may be OK.
The following minor points should be addressed:
11. Table S1 (supplement): With 21 IDC and 8 ILC, there are “1 Other” and “17 Unknown” breast cancer diagnoses, i.e. a mix of different subtypes of breast cancers, altogether a rather small cohort of 47, or four groups of 1, 8, 17, 21 – and likely not re-evaluated/confirmed by one or one group of pathologists (if so, this could be mentioned in the Materials and Methods), which may explain that “1 Other”, which appears to be known, since it is not in the group of “17 Unknown”.
22. The small circles within the figure should be explained somewhere. Statistically the authors confirm the effects claimed. The miR-200s also predicted the CTC status. Perhaps, one may discuss in a short note the possible origin of the miR-200 from CTCs?
33. Perhaps, one may discuss in a short sentence the possible (or not) origin of the miR-200 from CTCs?
44. L 89/90 check/correct: „human epidermal growth 2” –>
human epidermal growth factor receptor 2 (HER2)
55. Since the cohort is from 2010/2011 one may include a remark how future technology may increase sensitivity and specificity of miRs and CTCs, which may also influence OS and PFS?
Author Response

(The authors gave the same response as above.)
